# Components from the Human c-myb Transcriptional Regulation System Reactivate Epigenetically Repressed Transgenes

**DOI:** 10.3390/ijms21020530

**Published:** 2020-01-14

**Authors:** Cassandra M. Barrett, Reilly McCracken, Jacob Elmer, Karmella A. Haynes

**Affiliations:** 1School of Biological and Health Systems Engineering, Arizona State University, 501 East Tyler Mall, Tempe, AZ 85287, USA; cmbarre6@asu.edu; 2Department of Chemical Engineering, Villanova University, 217 White Hall, 800 East Lancaster Avenue, Villanova, PA 19085, USA; rmccrac1@villanova.edu (R.M.); jacob.elmer@villanova.edu (J.E.); 3Wallace H. Coulter Department of Biomedical Engineering, Emory University, Atlanta, GA 30322, USA

**Keywords:** MYB, c-myb, transgene, epigenetic silencing, activator, heterochromatin, polycomb

## Abstract

A persistent challenge for mammalian cell engineering is the undesirable epigenetic silencing of transgenes. Foreign DNA can be incorporated into closed chromatin before and after it has been integrated into a host cell’s genome. To identify elements that mitigate epigenetic silencing, we tested components from the c-myb and NF-kB transcriptional regulation systems in transiently transfected DNA and at chromosomally integrated transgenes in PC-3 and HEK 293 cells. DNA binding sites for MYB (c-myb) placed upstream of a minimal promoter enhanced expression from transiently transfected plasmid DNA. We targeted p65 and MYB fusion proteins to a chromosomal transgene, *UAS-Tk-luciferase*, that was silenced by ectopic Polycomb chromatin complexes. Transient expression of Gal4-MYB induced an activated state that resisted complete re-silencing. We used custom guide RNAs and dCas9-MYB to target MYB to different positions relative to the promoter and observed that transgene activation within ectopic Polycomb chromatin required proximity of dCas9-MYB to the transcriptional start site. Our report demonstrates the use of MYB in the context of the CRISPR-activation system, showing that DNA elements and fusion proteins derived from c-myb can mitigate epigenetic silencing to improve transgene expression in engineered cell lines.

## 1. Introduction

The advancement of cell engineering requires robust and reliable control of endogenous and synthetic genetic material within living cells. A lack of tools for enhancing the expression of transgenes in mammalian cells currently limits effective gene regulation in different contexts. Unpredictable formation of heterochromatin around transgenic material in mammalian cells limits our ability to express foreign DNA for the production of therapeutic proteins and the development of engineered mammalian systems for biosensing and computing [1,2]. Integrated transgenes are often silenced by the same mechanisms that serve as a cellular defense against viral insertion into the genome [3,4,5]. Nucleation of heterochromatin around transgenic material can be initiated and sustained by promoter methylation [1,5] and various histone modifications [2,4]. For example, MyD88 pathway-mediated silencing of transgenes leads to an accumulation of repressive H3K9me on newly bound histones [2,6]. Silencing of transgenes may also be Polycomb-mediated, where Polycomb repressive complexes deposit H3K27me3 on histones to establish a silenced state [7,8,9]. The frequency of undesired transgene silencing has led to the development of tools for mammalian cell engineering specifically aimed at combating heterochromatin.

Recruiting activators to a specific locus in order to reverse epigenetic silencing can be achieved either by including an activation-associated cis-regulatory DNA sequence within the construct itself, or through the targeting of engineered fusion proteins to the silenced transgene. Both natural and synthetic cis-regulatory motifs that recruit activators have been used [10,11,12,13] to help increase transgene expression as an alternative to viral promoters that are prone to methylation and silencing [1]. Previous screens by ourselves and other groups [11,14,15] have identified mammalian activation-associated cis-regulatory elements that recruit endogenous factors to increase the expression of epigenetically silenced transgenes, including motifs for nuclear factor Y, CTCF, and elongation factor alpha (EF1-α) [12,13]. The underlying regulatory mechanisms are not entirely understood, since in this case efficient screening for functional sequences has been prioritized over dissecting the mechanism of individual elements.

Fusion proteins that target activation-associated domains to transgenes can also be used to reverse silencing. Targeted epigenetic effectors such as p300 (histone acetyltransferase) and Tet1 (methylcytosine dioxygenase) are potent activators of gene expression [16,17,18]. These directly alter local chromatin features, therefore their function may be context dependent [17,18]. Transcriptional activation domain (TAD) proteins, including Herpes simplex virus protein vmw65 tetramer (4× VP16, VP64) and nuclear factor NF-kappa-B p65 subunit (p65) have been used singly, or as subunits within compound activators such as VPR, SAM, and SunTag [19,20,21]. Site-specific targeting of VP64 (4× VP16) enhances endogenous gene expression, and remodels chromatin through the accumulation of activation-associated histone modifications (H3K27ac and H3K4me) [20,22,23]. p65-based systems are also very effective at restoring both endogenous [19,24] and transgenic [25] gene expression. 

Significant progress towards transgene reactivation has been made so far, but several important gaps remain. First, several natural mechanisms of activation are not yet represented in published cell engineering studies. Chromatin remodelers that shift, remove, or exchange nucleosomes [26], and pioneer factors increase DNA accessibility in closed chromatin by displacing linker histones [26,27,28] remain under-utilized for transgene regulation. Second, the critical parameters for stable transgene activation are not yet fully defined. So far, at least two studies have demonstrated prolonged enhancement of transgenes (10 to 25 days) via targeted fusion proteins alone [29] or in combination with flanking anti-repressor DNA elements [30]. Neither study evaluated the chromatin features at the target genes prior to their reactivation, therefore the context in which expression enhancement occurred is uncertain. Finally, the performance of targeted activators can be context-dependent. Catalytic domains used for site-specific chromatin remodeling [18,30,31], may be inhibited by pre-existing chromatin features that vary across loci. For example, Cano-Rodriguez et al. constructed a targeted histone methyltransferase fusion and found that the endogenous chromatin microenvironment, including DNA methylation and H3K79me, impacted the ability of their fusion protein to deposit H3K4me and induce activation [32]. Similarly inconsistent performance has been shown for other fusions that generate H3K79me and H3K9me [33,34]. Systematic studies at loci with well-defined chromatin compositions are needed to fully understand mechanisms of chromatin state switching. 

Here, we expand our previous work where we had identified cis-regulatory sequences that enhanced expression from plasmid-borne transgenes [12]. To regulate expression of chromosomally-inserted transgenes, we built site-specific fusion proteins with effector modules that represent diverse activities: transcriptional activation through cofactor recruitment, direct histone modification, and nucleosome repositioning and displacement. We focus on reversal of silencing within Polycomb heterochromatin, which is known to accumulate at transgenes that are integrated into chromosomes [7,8,9] and is widely distributed across hundreds or thousands of endogenous mammalian genes that play critical roles in normal development and disease [9,35,36]. We report that recruitment of p65 and MYB-associated components via a cis-regulatory element or fusion proteins enhances expression from transgenes. MYB-mediated activation within transcriptionally repressive Polycomb heterochromatin relies on interactions with p300 and CBP. Our results have implications for determining the most appropriate strategy to enhance gene expression, specifically within Polycomb-repressed chromatin.

## 2. Results

### 2.1. Identification of Activation Associated Proteins

We surveyed public data to identify epigenetic enzymes and other proteins that are associated with transcriptional activation, and therefore might effectively disrupt repressive Polycomb chromatin. Polycomb-enriched chromatin typically includes Polycomb Repressive Complex 1 (PRC1: RING1A/B, PCGF1–PCGF6, CBX2, PHC1–PHC3, and SCMH1/2) [37], PRC2 (EZH1/2, EED, Suz12, and RBBP4/7) [37], H3K27me3, histone deacetylation, H2AK119ub1, and lncRNAs [37,38]. Each activation-associated protein (AAP) generates modifications of histone tails either through intrinsic catalytic activity or through the recruitment of chromatin-modifying co-factors. In order to predict how these AAPs might influence Polycomb heterochromatin, we searched the STRING protein-protein interaction database for binding partners and their associated chromatin-modifying activities (Appendix A).

The AAPs fall into six general categories. The transcriptional activation group, (NFkB)-p65 and the MYB (c-myb) transcriptional activation domain (TAD), includes proteins that recruit RNA Polymerase II (PolII) and p300/CBP, respectively. For comparison to a strong, commonly used activator we included the recombinant TAD VP64 (four tandem copies of VP16). These AAPs have no known intrinsic gene regulation activity, and rely upon the recruitment of co-activator proteins to stimulate transcription [39,40,41]. Histone modifications generated by the co-activators are primarily associated with a transcriptionally active state. 

The histone acetylation (HAT) group includes ATF2, P300, and KAT2B, all of which acetylate H3K27. P300 is associated with the recruitment of CBP and other co-activators that generate the activation associated mark H3K4me [42]. The histone H3 methyltransferase (H3 MT) group and the H4 methyltransferase (H4 MT) group include either Mixed-Lineage Leukemia (MLL) complex components or SET proteins. SETD7 deposits the activation associated modification H3K4me, but its regulatory impact may vary based on local DNA methylation, which can enhance or impede co-recruitment of repressive cofactors. The histone H4 methyltransferase PRMT5 induces histone acetylation that is associated with DNA methylation in some contexts [43]. Still, PRMT5 primarily acts as an activator.

The final two groups, chromatin remodelers (CR) and pioneer factors (PF) represent activities that are relatively underexplored in the design of fusion-protein regulators. SMARCA4 is a chromatin remodeler that relies on an ATP-dependent reaction to shift the position of nucleosomes at a target site [44]. It does not mediate the deposition of histone modifications, but is associated with CBP recruitment that evicts Polycomb-associated histone modifications [45]. PFs are represented in our library by FOXA1, a winged-helix protein that displaces linker histones from DNA to facilitate a transition to open chromatin [46]. In general, PFs bind to DNA within heterochromatin and do not catalyze histone post-translational modifications [28].

Several of the AAPs in our panel are associated with the eviction of Polycomb repressive complexes (PRCs) from endogenous genes. Accumulation of the chromatin remodeling protein SMARCA4 (BRG1) leads to the loss of PRCs at *Pou5f1* in mouse cells [47] and at *INK4b-ARF-INK4a* in human malignant rhabdoid tumor cells [48]. In the latter case, KMT2A (MLL1) also participates in PRC depletion. ATF2 interacts with a kinase that generates H3S28p, which antagonizes PRC binding [49,50,51]. Acetylation and methylation at H3K27 are mutually exclusive [52,53], therefore the AAPs associated with H3K27ac (p65, MYB, ATF2, P300, KAT2B) might contribute to PRC eviction (Appendix A). None of the AAPs in our panel are associated with enzymatic erasure of H3K27me3.

### 2.2. Cis-Regulatory Elements Recognized by Transcriptional Activators p65 and MYB Enhance Expression from an Extra-Chromosomal Transgene 

First, we used enhancer DNA elements to regulate expression from transiently-transfected plasmid DNA. Work from our group [54] and others [55,56] has shown that plasmid DNA becomes occupied by histones, which may contribute to transgene silencing in human cells. In a previous study, we used DNA sequences that were known targets of endogenous activation-associated proteins to reduce silencing of a *luciferase* reporter gene [12]. Here, we tested additional motifs (Figure 1A) that are recognized by AAPs from the transcriptional activator group in our panel, MYB and p65, in PC-3 cells. Compared to easy-to-transfect cell lines like HEK 293, prostate PC-3 cells have a lower transient transfection efficiency, e.g., about 50% EGFP-positive cells in samples treated with Lipofectamine/pEF-GFP in our hands, and a lower level of GFP or luciferase reporter expression. Therefore, we chose PC-3 to potentially observe a significant enhancing effect from the MYB and p65 motifs.

One of three MYB enhancer variants or the p65 enhancer was placed in either a forward (+) or reverse (−) orientation upstream of an EF1a promoter and a *luciferase* reporter (Figure 1B). PC-3 cells were transfected with each plasmid as described previously [12]. The highest mean levels of enhanced expression were observed for *MYB-G* − (3.4-fold, *p* = 8.6 × 10 ^−6^), *MYB-A* + (3.1-fold, *p* = 1.6 × 10 ^−6^), *MYB-C* − (2.7-fold, *p* = 3.1 × 10 ^−4^), and *MYB-A* − (2.3-fold, *p* = 5.0 × 10 ^−4^) (Figure 1C). For these constructs, Luc signal values of all individual replicates were higher than the mean control value. For the remaining MYB and p65 constructs, mean Luc signal values were roughly 2-fold higher than the negative control (*p* = 9.9 × 10 ^−3^ to 1.5 × 10 ^−2^), but some of the individual replicates were at or below the mean negative control value. Overall, these results suggest that certain cis-regulatory elements from the MYB system are potent enhancers that might attract endogenous transcriptional activators to drive transgene expression from a minimal promoter. 

### 2.3. Identification of Fusion Activators with Robust Activity within Polycomb Heterochromatin

Next, we asked whether the individual proteins MYB and p65, as well as other AAPs could enhance transgene expression in the absence of a specific enhancer sequence. To determine AAP activity within silenced chromatin, we targeted AAP fusion proteins (Appendix A) to a chromosomal luciferase reporter that had been previously targeted by Polycomb repressive complexes (PRCs). The AAP open reading frames (ORFs) encode catalytic subunits or full length proteins (Appendix A) that have been shown to support an epigenetically active state in several prior studies [39,40,44,46,58,59,60,61,62,63,64]. All of these ORFs exclude DNA binding and histone binding domains, except for the ORF encoding FOXA1, which has a catalytic domain that requires histone interactions. We cloned each ORF into mammalian vector 14 (MV14) (Appendix A) to express a Gal4-mCherry-AAP fusion. The Gal4 DNA binding domain serves as a module to target AAPs to upstream activation sequences (UAS) in the target transgene, while the mCherry tag allows for protein visualization and quantification of the activator fusion.

We tested all sixteen Gal4-AAP candidate fusion activators at a site that was enriched for ectopic Polycomb repressive complexes (PRCs) in HEK 293 (human embryonic kidney) cells. The HEK 293 cell line Gal4-EED/luc, carries a stably integrated firefly luciferase transgene with an upstream UAS (*Gal4UAS-Tk-luciferase*) (Figure 2A) [25,65]. The cells also carry a *TetO-CMV-Gal4EED* construct, which encodes a Gal4 DNA-binding domain (Gal4) fused to an embryonic ectoderm development (EED) open reading frame under the control of TetO-CMV promoter. The addition of doxycycline (dox) to cultured Gal4-EED/luc cells releases the TetR protein from *TetO-CMV-Gal4EED*, initiating the expression of Gal4-EED. Gal4-EED binds to the UAS site and recruits EZH2 (a PRC2 subunit that methylates histone H3 at lysine 27) to the reporter. Expression of *Tk-luciferase* is switched from active to silenced through the accumulation of polycomb chromatin components, which have been detected by chromatin immunoprecipitation (ChIP) experiments: EZH2, Suz12, CBX8, [65], and H3K27me3 [25,65] (Figure 2B). This well-characterized system allows us to test the activity of Gal4-AAPs with *a priori* knowledge of the chromatin environment at the target gene.

Gal4-EED/luc cells were treated with dox for two days to induce heterochromatin at the *Tk-luciferase* transgene. Afterwards, cells were grown for four days without dox to allow for Gal4-EED depletion. The four-day time point was chosen based on a previous report from Hansen et al. where PRC chromatin (CBX8 and H3K27me3) persisted after Gal4-EED levels had decreased (Figure 2B). Cells were then transfected with individual Gal4-AAP plasmids. *Luciferase* expression was measured three days after transfection. 

Three of the sixteen Gal4-AAP-expressing samples showed increased luciferase levels compared to a mock-transfected control (Lipofectamine reagent only) (*p* < 0.05) (Figure 2C). To investigate whether the other Gal4 fusions were inhibited by PRC chromatin, we tested the fusion proteins at open chromatin. We used a parental HEK 293 cell line, Luc14, that carries the *firefly luciferase* construct (*Gal4UAS-Tk-luciferase*) but lacks the *TetO-CMV-Gal4EED* repressor cassette [65]. Luciferase is constitutively expressed at intermediate levels in these cells. Again, we observed that only the three Gal4-AAP fusions from the transcriptional activation group stimulated expression when targeted to the promoter-proximal UAS site (Appendix A). In both chromatin states, transcriptional activation-associated AAP’s significantly increased expression compared to a mock transfection control (*p* < 0.05) by up to five-fold. Our results are consistent with other groups’ studies, where p65, VP64, or MYB stimulated gene expression from a promoter-proximal site [39,40,41]. Here, we have demonstrated activities of these proteins within PRC-enriched chromatin.

### 2.4. Gal4-MYB-Induced Activation at Tk-Luciferase Resists Complete re-Silencing over Time

The results so far were obtained at a single time point after Gal4-AAP expression. We were interested in determining whether transgene activation within polycomb chromatin is stable or is transient and susceptible to eventual re-silencing [66]. To investigate this question, we performed time-course experiments to measure expression from re-activated *luciferase* over time (Figure 3A). We induced Polycomb heterochromatin in Gal4-EED/luc cells as described for the previous experiments. Two days after transfection with one of the activators, Gal4-VP64, -P65, or -MYB, cells were grown in dox-free medium supplemented with 10 μg/mL puromycin to select for Gal4-AAP positive cells. After three days of selection, we measured *luciferase* expression, Gal4-AAP mRNA levels, and mCherry fluorescence from a sample of each culture. The cells were then passaged into puromycin-free, dox-free medium to allow for the loss of Gal4-AAP, sampled every four days (approximately three cell divisions), and the same three measurements (luciferase, Gal4-AAP mRNA, and mCherry) were repeated. 

For all three Gal4-AAP fusions, *luciferase* expression was significantly elevated at most time points (*p* < 0.05) compared to a mock transfection control (Lipofectamine reagent only) (Figure 3B). Steep declines of Gal4-AAP mRNA and mCherry fluorescence after three days (Figure 3C,D) confirmed that the activators were transiently expressed and then depleted. Therefore, enhanced gene expression persisted to varying degrees after depletion of each Gal4-AAP, suggesting epigenetic memory of *luciferase* activation. Fluctuations in *Tk-luciferase* expression over cell culture passages suggest that the activated state became unstable after depletion of the transactivator. After we ended selection for Gal4-AAP expression, Luc signal levels decreased roughly 3-fold from day 3 to day 7 (Gal4-VP64 3.3-fold, Gal4-P65 3.8-fold, Gal4-MYB 2.9-fold, *p* < 0.01), but remained slightly and significantly higher (*p* < 0.05) than repressed levels (control) in Gal4-VP64 and Gal4-MYB cells (Figure 3B). Luc signals spiked at day 11 (Gal4-P65 4.3-fold, *p* < 0.01) or day 15 (Gal4-VP64 1.7-fold, Gal4-MYB 1.8-fold, *p* < 0.01), and then decreased by day 19 (Gal4-VP64 2.5-fold, Gal4-P65 2.2-fold, Gal4-MYB 1.6-fold, *p* < 0.01). We observed similar fluctuations in luciferase expression in an additional trial (Appendix A). This instability may be caused by competition between the activated state and background levels of Gal4-EED activity, which we have previously observed as weak levels of repression prior to dox treatment of Gal4-EED/luc cells [25]. Overall, the Gal4-MYB-activated *Tk-luciferase* transgene showed the most resistance to re-silencing. In this case, expression levels remained 1.6-fold or higher than the negative control for up to 19 days in one trial, and up to 15 days in an additional trial.

### 2.5. MYB-Mediated Activation within Closed Chromatin Requires Interaction with a Histone Acetyltransferase

Next, we used a chemical inhibitor to probe the mechanism of MYB-driven enhancement of gene expression. The TAD core acidic domain of human MYB (D286-L309) included in our Gal4-MYB fusion construct is known to interact with a protein heterodimer of p300 and CBP (Appendix A). A single base pair mutation within the MYB TAD domain (M303V) disrupts p300 recruitment and subsequent activation by MYB indicating that this recruitment is crucial to activation by MYB [67,68]. The p300/CBP histone acetylation complex deposits H3K27ac in opposition to H3K27me3 induced by PRC2 [69,70]. Therefore, Gal4-MYB-induced activation within Polycomb heterochromatin may be driven by histone acetylation.

To test this idea, we treated cells with a compound known to disrupt the activity of the MYB/p300/CBP complex. Celastrol is a minimally toxic pentacyclic triterpenoid that directly inhibits the MYB/p300 interaction, by binding to the KIX-domain of CBP which serves as a docking site for the formation of the MYB/p300/CBP complex [71,72,73,74] (Figure 4A). Gal4-EED/luc cells were treated with dox to induce polycomb chromatin and then transfected with Gal4-MYB as described for previous experiments. We treated these cells with 5 μM celastrol for six hours. MTT assays indicated no toxicity at this concentration (Appendix A). We observed a significant (*p* < 0.05) decrease in *luciferase* expression in celastrol-treated cells compared to an untreated control (Figure 4B). This result suggests that Gal4-MYB activity requires an interaction between MYB and p300/CBP. The other two activators, Gal4-VP64 and Gal4-P65, were insensitive to celastrol (Figure 4B), indicating a p300/CBP-independent mechanism for these two fusions. 

In a time-course experiment, we observed that Gal4-MYB activity can be switched by adding or removing celastrol from the growth medium. Eighteen hours after removal of celastrol from Gal4-MYB-treated cells, *luciferase* expression levels increased significantly (*p* < 0.05 compared to repression at t = 6 h.) (Figure 4C). Re-addition of celastrol led to a reduction of Gal4-MYB-induced expression.

### 2.6. MYB-Mediated Activation in Polycomb Heterochromatin Relies upon Proximity to the Transcriptional Start Site

Next, we asked whether MYB-mediated activation at transgenes is context dependent. We leveraged the flexible dCas9/sgRNA system to target the MYB TAD to several sites along the *Tk-luciferase* transgene (Appendix A). We induced Polycomb heterochromatin in HEK 293 Gal4-EED/luc cells with dox, then removed dox to allow for Gal4-EED depletion as described above. We transfected the cells with one of four dCas9-MYB constructs, each carrying a different sgRNA targeted within the *luciferase* transgene. After three days, we determined *luciferase* expression compared to mock-transfected control cells (Lipofectamine reagent only). In cells where dCas9-MYB was targeted closest to the transcription start site (+9 bp) *Tk-luciferase* expression reached the levels we observed for Gal4-MYB bound to the yeast Gal4 upstream activation sequence (UAS) (Figure 5). Expression enhancement from downstream target sites was significantly lower than Gal4-MYB (*p* < 0.05), suggesting position-dependent activity at the model Polycomb-repressed locus.

We also tested MYB at a green fluorescent protein (GFP) transgene that had become transcriptionally downregulated, presumably by endogenous heterochromatin. The construct, *GFP* under the control of a *CMV* promoter, had been inserted via Cas9-mediated HDR into a non-protein-coding region of the HEK 293 genome (HEK 293 site 3 [75]). After ten passages, the frequency of GFP-positive cells decreased from ~50% to ~2% (Appendix A). We transfected the cells with dCas9-MYB-expressing plasmids, and three days later we used flow cytometry to measure GFP fluorescence compared to a mock-transfected control (Lipofectamine reagent only). We observed a very small increase (~0.05%, *p* < 0.05) in the frequency of GFP-positive cells when dCas9-MYB was targeted near the CMV promoter at sgRNA site L3 (Appendix A), suggesting that dCas9-MYB was not sufficient to fully restore transgene expression in this case.

## 3. Discussion

We have demonstrated that DNA enhancer elements and fusion proteins derived from endogenous mammalian systems can be used to enhance expression from transiently transfected DNA. Furthermore, we have demonstrated that transient expression of Gal4-MYB confers some resistance to full re-silencing of a transgene in ectopic Polycomb heterochromatin. The artificial repressor (Gal4-EED) used for this study or the incomplete erasure of certain repressive chromatin marks may have caused instability of the activated state. Future work to map chromatin features of artificially activated states over time will shed light on the requirements for stable activation. So far, our results represent some progress towards achieving reliable expression of synthetic DNA in engineered cells.

Our results also suggest that reactivation of a transgene within Polycomb heterochromatin is supported by the recruitment of transcription initiation complexes. However, the precise chromatin remodeling mechanism is unclear since our STRING analyses did not reveal an obvious pattern of histone modifications to distinguish the ineffective Gal4-AAPs from activators that enhanced expression in Polycomb heterochromatin (Appendix A). Upon further investigation, we determined that assembly of the MYB TAD with P300/CBP is critical for Gal4-MYB-mediated activation within Polycomb chromatin. Celastrol inhibits the interaction of p300/CBP with MYB by binding to the CBP KIX domain [71,72,73,74], and completely reduces Gal4-MYB activity. In contrast, the Gal4-P65 and Gal4-VP64 fusions showed robust activation of PRC-silenced luciferase in the presence of celastrol (Figure 5). Although VP64 and p65 are known to interact with p300/CBP, they also interact with the large multi-subunit Mediator complex to initiate transcription [76,77,78]. Multiple interactions of Gal4-P65 and Gal4-VP64 with Mediator may allow these proteins to function independently of p300/CBP [79]. However in the case of Gal4-MYB, cooperative interactions between p300/CBP and Mediator [80,81] may be necessary for gene activation. Mediator is known to cooperatively counteract PRC2 repression [82] and certain Mediator subunits are directly involved in the removal of PRC2 from endogenous promoters [83]. Furthermore, Mediator is an antagonist of the PRC1 repression complex [84].

The inhibitor experiments also demonstrate a new technique for chemically-inducible gene regulation in mammalian cells. The ability to quickly toggle between enhanced and repressed states is a fundamental feature of engineered transgenic systems [29,85,86]. Current methods for toggling gene expression in mammalian cells employ drug-mediated transactivator localization, such as allosteric modulation of DNA-binding protein domains [29,85,87], blue light-responsive CRY proteins [88], and chemically induced dimerization (CID) systems [89,90,91], or RNA interference to deplete the regulator [86]. To our knowledge, no systems currently exist where the transactivation module’s activity (i.e., MYB-CBP binding) is modulated by a small molecule drug. Celastrol has low toxicity and is in fact being explored as a therapeutic due to its positive effects on the immune system [92,93,94]. The concentration of celastrol that is sufficient to toggle Gal4-MYB activity in polycomb chromatin is well below the reported LD50 values for this drug [95,96,97,98,99].

Finally, our work demonstrates the potential flexibility of MYB fusion proteins as transactivators. We demonstrated targeted reactivation of a transgene using either a Gal4-MYB or a dCas9-MYB fusion protein. However, the results also suggest limitations to the use of MYB such as a requirement for TSS-proximal positioning as indicated by the results from targeting dCas9-MYB to *Tk-luciferase*. Furthermore, MYB may not be effective or may require additional activating factors to stimulate transcription at other genes, such as the CMV-GFP transgene we tested here. Factors that might account for the different responses of transgenes to MYB include intrinsic differences in the core promoter sequences and differences in chromatin structure. 

In conclusion, our study showed that placing DNA binding sites for MYB (c-myb) upstream of a minimal promoter enhances expression from transiently transfected plasmid DNA in prostate PC-3 cells. We also showed that the core transcriptional activation domain (TAD) from the MYB protein activates expression from chromosomal transgenes had been previously silenced by ectopic Polycomb complexes. Finally, we showed that the activity of a MYB fusion protein can be reversibly switched off and on by the addition or removal of a non-toxic concentration of celastrol. These results demonstrate that DNA elements and fusion proteins derived from c-myb can be used to mitigate epigenetic silencing and to regulate gene expression in genetically engineered human cell lines.

## 4. Materials and Methods

### 4.1. Construction and Testing of Plasmids Containing MYB- and p65 Motifs

Plasmid construction, transfection of PC-3 cells, and luciferase assays were carried out as described previously [12]. Briefly, cloning of double-stranded oligos was used to insert motifs 222 bp upstream of the transcription start site of an EF1a promoter at XbaI/SpeI. Plasmids were then transfected into PC-3 cells (ATCC, CRL-1435) using Lipofectamine LTX™ following the manufacturer’s recommended protocols. Luciferase expression was measured 48 h after transfection using a luciferase assay kit (Promega, Madison, WI). All luciferase values were normalized relative to the native plasmid control, which contained an unaltered EF1a promoter.

### 4.2. Construction of MV14 and Gal4-AAP Plasmids

We constructed mammalian expression vector 14 (MV14) for the overexpression of Gal4-mCherry-AAP fusion proteins in-frame with a nuclear localization sequence and 6X-histidine tag. First, plasmid MV13 was built by inserting a Gal4-mCherry fragment into MV10 [100] directly downstream of the CMV promoter. Next, MV14 was built by inserting a SpeI/PstlI (FastDigest enzymes, ThermoFisher Scientific) -digested gBlock Gene Fragment (Integrated DNA Technologies), which encoded a XbaI/NotI multiple cloning site, into MV13 downstream of mCherry. Ligation reactions included gel-purified (Sigma NA1111) DNA (25 ng linearized vector, a 2× molar ratio of insert fragments), 1× Roche RaPID ligation buffer, and 1.0 uL T4 ligase (New England Biolabs), in a final volume of 10uL.

AAPs were cloned into MV14 at the multiple cloning site containing XbaI and NotI cut sites. AAPs were either ordered from DNASU in vectors and amplified using primers that added a 5′ XbaI site and a 3′ NotI site or ordered as gBlock Gene Fragments with the same 5′ and 3′ cutsites (Integrated DNA Technologies). Sequences in vectors were amplified with Phusion High Fidelity DNA Polymerase (New England BioLabs) and primers listed in Appendix A. MV14 and AAP inserts were double-digested with FastDigest *XbaI* and FastDigest *NotI* (ThermoFisher Scientific) and then ligated with T4 DNA ligase (New England Biolabs). MV14_AAP plasmids are publicly available through DNASU (Appendix A).

### 4.3. Cell Culturing and Transfections

Luc14 and Gal4-EED/luc HEK 293 cells were grown in Gibco DMEM high glucose 1× (Life Technologies) with 10% Tet-free Fetal Bovine Serum (FBS) (Omega Scientific), 1% penicillin streptomycin (ATCC) at 37 °C in a humidified 5% CO_2_ incubator. Gal4-EED/luc cells were treated with 1 µg/mL doxycycline (Santa Cruz Biotechnology) for two days to induce stable polycomb repression. Dox was removed and cells were cultured for another four days before being seeded in 12-well plates. Luc14 cells and dox-induced Gal4-EED/luc cells were seeded in 12-well plates such that cells reached 90% confluency for lipid-mediated transfection. Transfections were performed with 1 µg plasmid per well, 3 µL Lipofectamine LTX, and 1 µL Plus Reagent (Life Technologies) per the manufacturer’s protocol. Seventy-two hours post transfection, cells were either collected for analysis or passaged further.

Puromycin selection was carried out on Gal4-AAP-expressing cells for the experiments represented in Figure 5 and Appendix A. Dox-treated Gal4-EED/luc cells were transfected in 12-well plates and then grown for 24 h before the addition of 10 µg/mL puromycin (Santa Cruz Biotechnology) to Gibco DMEM high glucose 1× (Life Technologies) with 10% Tet-free Fetal Bovine Serum (FBS) (Omega Scientific), 1% penicillin streptomycin (ATCC). Cells were grown in puromycin containing media for two days before wash out.

### 4.4. Luciferase Assays

Luciferase assays were performed as previously described in Tekel et al. [100]. In brief, a single well of cells from a 12 well tissue culture plate was collected per independent transfection in 1.5mL 1× PBS. Cells were loaded into nine wells of a Black Costar Clear Bottom 96 Well Plates (Corning #3631). Three wells of cells were used to detect mCherry in order to quantify Gal4-AAP proteins. A 2× Hoechst 33,342 stain (Invitrogen #H3570) was loaded into three more wells to stain nuclear DNA in order to quantify cell density. The final three wells were prepared with Luciferase Assay Buffer (Biotium #30085). Plates were scanned in a microplate reader (Biotek Synergy H1) to detect RFP (580–610 nm), Hoechst 33,342 fluorescence (360–460 nm) and chemiluminescence from the same sample in parallel.

### 4.5. RT-qPCR

We prepared total RNA from ~1.0 × 10^6^ cells (Qiagen RNeasy Mini kit 74104) and generated cDNA from 2 µg of total RNA and the SuperScript III First Strand Synthesis system (Invitrogen #18080051) in a reaction volume of 20 μL. Quantitative PCR (qPCR) was performed with universal primers against the mCherry portion of the Gal4-AAP fusions, or the *TATA binding protein* (*TBP*) housekeeping gene. Triplicate qPCR reactions (10 μL) each contained SYBR Green 1 2× master mix (Roche), 2 µL of a 1:10 cDNA dilution, and 750 nM of each primer (forward and reverse, see Appendix A). We calculated Mean Quantification Cycle (C_q_) for three replicate wells per unique reaction. Change in gene expression level was calculated as ΔC_q_ = 2[Mean Cp reference − Mean Cp target]. Log2 fold change in gene expression was calculated as = log2(ΔC_q transfected cells_/ΔC_q mock_). 

### 4.6. Flow Cytometry

Cells were passed through a 35 μm nylon strainer (EMS #64750-25). Green fluorescent signal from GFP and red fluorescent signal from mCherry were detected on a BD Accuri C6 flow cytometer (675 nm LP filter) using CFlow Plus software. Data were further analyzed using FlowJo 10.6.1. One run (∼10,000 live cells, gated by forward and side scatter) was completed per sample, allowing us to determine median fluorescence within the live cell population. 

### 4.7. Construction of dCas9-MYB and Design of sgRNAs

We modified the vector pX330A_dCas9–1 × 4 (a gift from Takashi Yamamoto, Addgene plasmid #63598) by inserting a gBlock Gene Fragment (Integrated DNA Technologies) encoding the MYB TAD followed by a p2A signal [101] and *mCherry* after the *dCas9* ORF. The resulting vector expresses a dCas9-MYB fusion and mCherry as separate proteins from a single mRNA transcript. The vector and gBlock were digested with *FseI* (New England BioLabs) and FastDigest *Eco*RI (ThermoFisher Scientific) and ligated using T4 DNA Ligase (New England BioLabs). We named this new vector pX330g_dCas9-MYB. SgRNAs used in the study (Appendix A) were designed using the CRISPR design tool at crispr.mit.edu. DNA oligos were synthesized with BbsI overhangs for cloning into pX330g_dCas9-MYB (Integrative DNA Technology). Drop-in of sgRNAs followed the cloning protocol described in Cong et al. [102]. 

### 4.8. Celastrol Treatments

Gal4-EED/luc cells were induced with dox and transfected as described above. Three days after transfection, cells were treated with Celastrol (Selleck Chemicals) at a final concentration of 5 μM in Gibco DMEM high glucose 1× (Life Technologies) with 10% Tet-free Fetal Bovine Serum (FBS) (Omega Scientific). Cells were incubated with the drug for six hours before being washed and either harvested for a luciferase assay or grown further in drug-free media. 

### 4.9. Statistical Analyses

The differences of means were calculated using the two sample, one-tailed Student’s *t* test. For *p* < 0.05, confidence was 95% for 2 degrees of freedom and a test statistic of *t*_(0.05,2)_ = 2.920. To evaluate the significance of Gal4-MYB induced activation after the removal of celastrol and its subsequent re-addition, a nested one-way ANOVA was used with 95% confidence and two degrees of freedom. 

## Figures and Tables

**Figure 1 ijms-21-00530-f001:**
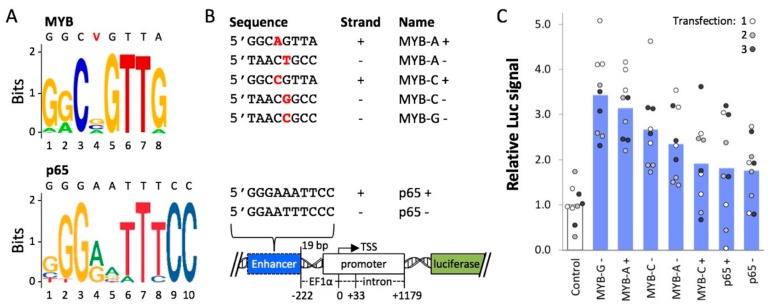
Luciferase expression from MYB- and p65-enhancer constructs. (**A**) Enhancer motif logos for MYB and p65 were generated by JASPAR [57]. The MYB sequence includes a variable site (V) equally represented by A, C, or G nucleotides, shown in red font. (**B**) *Luciferase* reporter constructs included one of the enhancer sequences (MYB-A+, etc.) 19 bp upstream of an EF1α promoter, or no enhancer (Control). (**C**) Luciferase assays were carried out using PC-3 cells transfected with Lipofectamine-plasmid complexes. For each transfection, luminescence (Luc signal) values were measured in triplicate and normalized to the average signal from the Control. Circle = one Luc measurement, bar = mean of nine Luc values.

**Figure 2 ijms-21-00530-f002:**
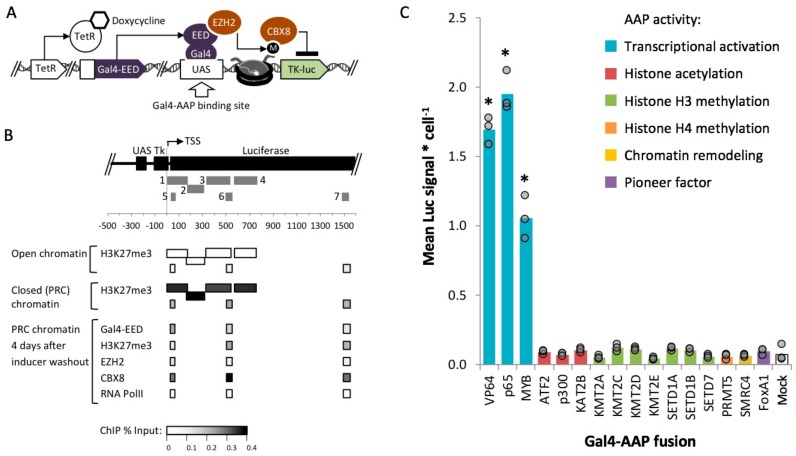
Measurement of *luciferase* reporter expression within closed or open chromatin after exposure to Gal4-AAP fusions. (**A**) In Gal4-EED/luc HEK 293 cells expression of the Gal4-EED fusion protein is controlled by a Tetracycline repressor (TetR). Treatment with dox allows expression of Gal4-EED, which binds UAS and recruits EZH2 (a subunit of PRC2). EZH2 methylates (M) histone H3K27, which recruits CBX8 (a subunit of PRC1) (**B**) Panel B summarizes published chromatin immunoprecipitation (ChIP) data from previous analyses of the *Tk-luciferase* locus. Grey numbered bars indicate amplicons for quantitative PCR: 1-4 [25], and 5-7 [65]. (**C**) Dox-treated cells were transfected with each Gal4-AAP fusion plasmid. Three days after transfection, the luciferase signal was measured. Each circle in the bar graph shows the mean luciferase (Luc) signal for a single transfection, divided by cell density (total DNA, Hoechst staining signal). Bars show means of three transfections. Asterisks (*) = *p* < 0.05 compared to mock-transfected cells.

**Figure 3 ijms-21-00530-f003:**
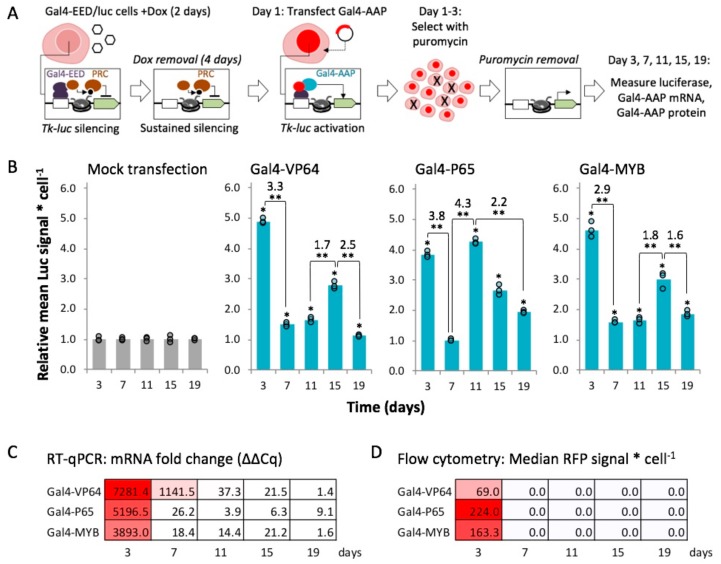
Expression of Polycomb-repressed *Tk-luciferase* over time after expression and loss of Gal4-P65, Gal4-VP64, or Gal4-MYB. (**A**) Diagram of the experimental workflow. Gal4-EED/luc cells were treated with dox to induce polycomb chromatin, transfected with a Gal4-AAP plasmid, and grown under puromycin selection (10 µg/mL). At three days post transfection, cells were sampled for assays, passaged in puromycin-free medium, then sampled seven, 11, 15, and 19 days post transfection for additional assays. (**B**) Individual values (circles, Luc signal per cell) at each time point are normalized by the mean of the mock-transfected (Lipofectamine-only) negative control. Fold change of mean values between time points for each Gal4-AAP experiment are shown within each bar graph. Asterisks represent *p*-values (* *p* < 0.05) for mean values greater than the mean for the mock-transfected negative control sample. * *p* < 0.05 and ** *p* < 0.01 are shown for intra-group comparisons (brackets). Results from an additional trial are shown in Appendix A. (**C**) Reverse transcription followed by quantitative PCR (RT-qPCR) with primers for the universal mCherry region was used to determine Gal4-AAP transcript levels. “mRNA fold change” represents the Cq value normalized by the Cq of a housekeeping gene (*TBP*), and relative to mock-transfected cells (Lipofectamine reagent only), log2 transformed. (**D**) Flow cytometry of mCherry signal (red fluorescent protein, RFP) was used to determine Gal4-AAP protein levels. Data in C and D were generated from one set of transfections in B. For other samples, cells were visually inspected for RFP to verify the loss of Gal4-AAP. In the tables in C and D, the intensity of red shading corresponds with the values shown in each cell.

**Figure 4 ijms-21-00530-f004:**
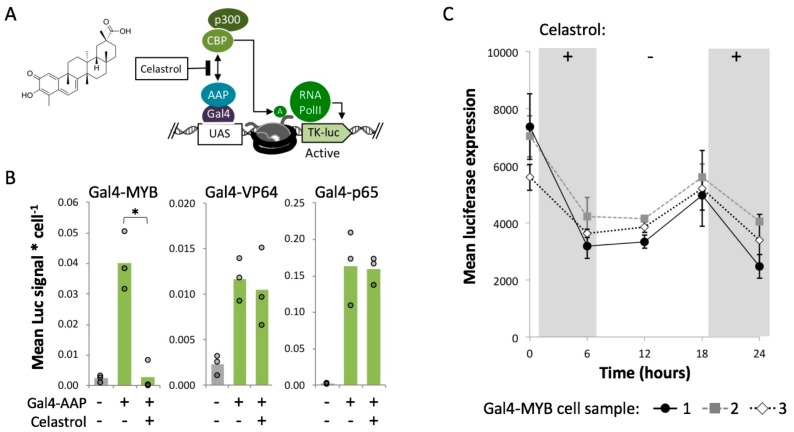
Celastrol disrupts Gal4-MYB-mediated activation of *luciferase* in closed chromatin. (**A**) The p300/CBP complex acetylates histones via the catalytic HAT domain of p300 and/or CBP [70]. Celastrol inhibits the recruitment of p300/CBP by MYB by binding a docking domain in CBP that facilities complex assembly [72,74]. (**B**) Three days after Gal4-EED-mediated repression of *Tk-luciferase* and transfection with Gal4-AAPs, cells were treated with 5 μM celastrol for six hours and collected for luciferase assays. Mean luciferase (Luc) signal per cell is presented as described for Figure 2C. Asterisk (*) = *p* < 0.05. (**C**) Luc measurements were carried out in Gal4-MYB-expressing cells after removal (−) and re-addition (+) of celastrol. Each series (Gal4-MYB cell sample) represents an independent transfection. Point = mean of three luciferase assays, bars = standard error.

**Figure 5 ijms-21-00530-f005:**
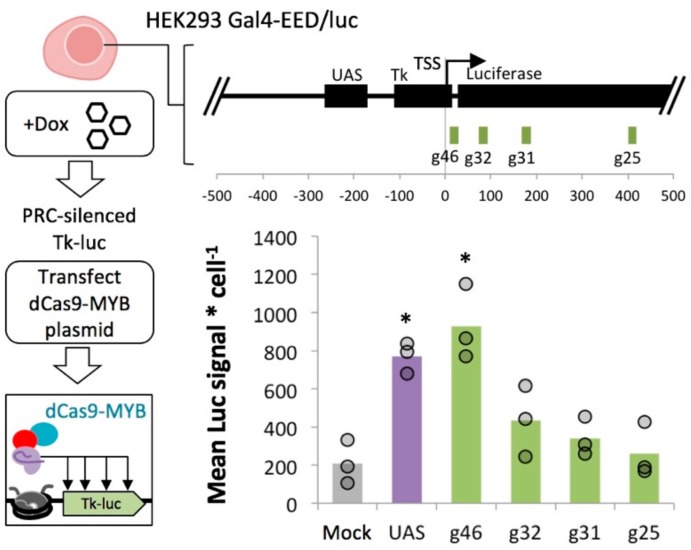
dCas9-MYB’s ability to enhance expression in induced Polycomb heterochromatin is dependent upon distance from the promoter. We targeted dCas9-MYB to four locations (g46, g32, g31, g25) across the *Tk-luciferase* transgene in silenced Gal4-EED/luc cells. Luciferase signal per cell is presented as described for Figure 2C. The negative control (grey bar) is a mock-transfection with Lipofectamine. The positive control (purple bar) is a transfection with Gal4-MYB, which binds the yeast upstream activation sequence (*UAS*) upstream of the *Tk* promoter. Asterisks (*) = *p* < 0.05 for experimental mean compared to the mock-transfected control mean.

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
