# Peer review of "Components from the Human c-myb Transcriptional Regulation System Reactivate Epigenetically Repressed Transgenes"

_ijms, 2020, doi:10.3390/ijms21020530_

Round 1
Reviewer 1 Report
In this study the authors identify elements involved in the reduction of epigenetic silencing of transgenes, as well as develop a novel approach to improve the expression of the transiently transfected DNA.
As the authors indicate, the current limits of the methodology applied to the expression of transgenes due to the cells defense mechanisms, it is one of the big challenges that we have to face in the cell engineering. Barrett et al., demonstrate with an appropriate methodology and a clear presentation how MYB components can epigenetically reactivate repressed transgenes through a cis-regulatory element or fusion proteins as transactivators.
This study is well design and the conclusions are supported by the results. Minor revisions have to be considered for its publication.
- Page 10, line 254: “Gal4-MYB-induced activation at TK-luciferase resist complete re-silencing over time”. It is not clear if this title is a new section or it should correspond to the point 2.4 of results. Modify this accordingly.
- Page 11, lines 259 to 267: The experimental description is a little confusing for this experiment. It is full indicated in the text, but it will help include an scheme of the experimental design in the Figure 3.
- Figure 3A & Results: Did the authors perform an analysis of the data intra-group for the each Gal4 transfected group over the time-course (VP64, P65, MYB)? As the authors indicate in the results, fluctuations were detected luciferase signal over cell-culture passages that should be consider in detail.
- Figure 3 B & C: Include the color legend for the tables in Figure 3 as well as indicate it in the results.
- Figure 4C: Number of samples included in the time-course experiment using the Celastrol. Also, Include the figure legend each group represented in this figure.
- Figure 5: same as for figure 3. Include a small scheme about the followed experimental design. It is well described in the text, but it will help to avoid confusions.
- Two types of cells were used in this study: PC-3 and HEK293. Include the reasons why the experiments were performed in different cells.
- Page 20, lines 508 to 514: Review the font.
- Discussion: Several conclusions can be extracted from this work and these are clearly indicated in the discussion. However, a small paragraph that summarizes all of them will be helpful to provide a general view of the study.
Author Response
Dear Reviewer 1,
We thank you for the time you took to read our manuscript, and for your insightful and constructive feedback. We hope the following responses and revisions adequately address your critiques.
Reviewer 1: Page 10, line 254: “Gal4-MYB-induced activation at TK-luciferase resist complete re-silencing over time”. It is not clear if this title is a new section or it should correspond to the point 2.4 of results. Modify this accordingly.
Response: This is meant to be a new section. We have corrected this formatting error, and all subsequent headers in the Results section have been renumbered.
Reviewer 1: Page 11, lines 259 to 267: The experimental description is a little confusing for this experiment. It is full indicated in the text, but it will help include an scheme of the experimental design in the Figure 3.
Response: We have added illustrations to Figure 3 to show key steps in the experiment as they relate to the text and to the time points in the bar charts.
Reviewer 1: Figure 3A & Results: Did the authors perform an analysis of the data intra-group for the each Gal4 transfected group over the time-course (VP64, P65, MYB)? As the authors indicate in the results, fluctuations were detected luciferase signal over cell-culture passages that should be consider in detail.
Response: We have included new intra-group analyses (t tests) for each set of Gal4-AAP luciferase expression data. We performed t tests to support our conclusions as follows: Text at page 11, line 276 has been changed to “After we ended selection for Gal4-AAP expression, Luc signal levels decreased roughly 3-fold from day 3 to day 7 (Gal4-VP64 3.3-fold, Gal4-P65 3.8-fold, Gal4-MYB 2.9-fold, p < 0.01), but remained slightly and significantly higher (p < 0.05) than repressed levels (control) in Gal4-VP64 and Gal4-MYB cells (Figure 3B).” Text at page 11, line 278 has been changed to “Luc signals spiked at day 11 (Gal4-P65 4.3-fold, p < 0.01) or day 15 (Gal4-VP64 1.7-fold, Gal4-MYB 1.8-fold, p < 0.01), and then decreased by day 19 (Gal4-VP64 2.5-fold, Gal4-P65 2.2-fold, Gal4-MYB 1.6-fold, p < 0.01).” In Figure 3, we added the fold-change values and p-values mentioned in the text, brackets to illustrate the intra-group comparisons, and added the following to the figure legend: “Fold change values between time points for each Gal4-AAP experiment are shown within each bar graph. Asterisks represent p-values (* p < 0.05) for mean values greater than the mean for the mock-transfected negative control sample. * p < 0.05 and ** p < 0.01 are shown for intra-group comparisons (brackets).” In Supplemental Figure S4, we performed the same comparisons for Trial 2.
Reviewer 1: Figure 3 B & C: Include the color legend for the tables in Figure 3 as well as indicate it in the results.
Response: The colors are scaled as a visual aid so that differences in the numerical values (shown within the boxes) are more intuitive to the reader. A color scale bar is not needed in this case.
Reviewer 1: Figure 4C: Number of samples included in the time-course experiment using the Celastrol. Also, Include the figure legend each group represented in this figure.
Response: Regarding the number of samples represented by each point, the current figure legend states that “Point = mean of three luciferase assays.” We have added a new legend to emphasize that each series represents an independent transfection of Gal4-MYB-expressing plasmids.
Reviewer 1: Figure 5: same as for figure 3. Include a small scheme about the followed experimental design. It is well described in the text, but it will help to avoid confusions.
Response: We have added a new panel A to illustrate the experimental design. We have removed the plasmid map from Figure 5 and moved it to Supplementary Figure S5.
Reviewer 1: Two types of cells were used in this study: PC-3 and HEK293. Include the reasons why the experiments were performed in different cells.
Response: PC-3 cells have a lower transient transfection efficiency, e.g. about 50% EGFP-positive cells in samples treated with Lipofectamine/pEF-GFP in our hands, and a lower level of GFP or Luciferase expression than easy-to-transfect cell lines like HEK 293. Therefore, we chose PC-3 to potentially observe a significant enhancing effect from the various MYB and p65 motifs. We have added this explanation to the Results (end of the first paragraph in section 2.2, line 69). Response: PC-3 is a widely-used cancer cell line that represents one of the deadliest carcinomas (prostate). The expression-enhancing MYB motif would be potentially useful in gene therapies against prostate cancer. We have added this point to the Discussion.
Reviewer 1: Page 20, lines 508 to 514: Review the font.
Response: We have changed the font style so that it is consistent with the entire manuscript.
Reviewer 1: Discussion: Several conclusions can be extracted from this work and these are clearly indicated in the discussion. However, a small paragraph that summarizes all of them will be helpful to provide a general view of the study.
Response: We have added the following new paragraph to the end of the Discussion section: “In conclusion, our study showed that placing DNA binding sites for MYB (c-myb) upstream of a minimal promoter enhances expression from transiently transfected plasmid DNA in prostate PC-3 cells. We also showed that the core transcriptional activation domain (TAD) from the MYB protein activates expression from chromosomal transgenes had been previously silenced by ectopic Polycomb complexes or by uncharacterized endogenous chromatin. Finally, we showed that the activity of a MYB fusion protein can be reversibly switched off and on by the addition or removal of a non-toxic concentration of celastrol. The results demonstrate that DNA elements and fusion proteins derived from c-myb can be used to mitigate epigenetic silencing and to regulate gene expression in genetically engineered human cell lines.”
Reviewer 2 Report
Overall, the manuscript is well-prepared and hardly can be improved without increase of page numbers.
Authors report on a very important methodological point an use a comprehensive array of tool to demonstrate that c-MYB is a vivid example of transcriptional regulator that manages to re-induce expression of reporter transgenes that were repressed by epigenetic mechanisms.
I see the paper of high merit and interest to a Reader.
Yours, Reviewer.
Author Response
Dear Reviewer 2,
Reviewer 2: “Overall, the manuscript is well-prepared and hardly can be improved without increase of page numbers.”
Response: We appreciate your very positive comments! Thank you for taking the time to review our paper.
Round 2
Reviewer 1 Report
I would really like to thank to the authors the time dedicated to address all the questions suggested in the first revision. I have no more comments to add to this version of the paper.
Author Response
Dear Reviewer 1,
Reviewer 1: “I would really like to thank to the authors the time dedicated to address all the questions suggested in the first revision. I have no more comments to add to this version of the paper.”
Response: We appreciate your very careful review and re-review. Thank you.